# Yield and Sensorial and Nutritional Quality of Strawberry (*Fragaria* × *ananassa* Duch.) Fruits from Plants Grown Under Different Amounts of Irrigation in Soilless Cultivation

**DOI:** 10.3390/plants14020286

**Published:** 2025-01-20

**Authors:** Davide Raffaelli, Rohullah Qaderi, Luca Mazzoni, Bruno Mezzetti, Franco Capocasa

**Affiliations:** Department of Agricultural, Food and Environmental Sciences, Università Politecnica delle Marche (UNIVPM), Via Brecce Bianche 10, 60131 Ancona, Italy; d.raffaelli@pm.univpm.it (D.R.); r.qaderi@staff.univpm.it (R.Q.); l.mazzoni@staff.univpm.it (L.M.); b.mezzetti@staff.univpm.it (B.M.)

**Keywords:** *Fragaria* × *ananassa*, strawberry, soilless, nutritional quality, yield, water management

## Abstract

Water scarcity is an ecological issue affecting over 10% of Europe. It is intensified by rising temperatures, leading to greater evaporation and reduced precipitation. Agriculture has been confirmed as the sector accounting for the highest water consumption globally, and it faces significant challenges relating to drought, impacting crop yields and food security. Sustainable practices, precision irrigation, and the development of drought-resistant crops are essential for the mitigation of this threat. Effective, innovative solutions are crucial for optimizing water use for intensive crops such as cultivated strawberries (*Fragaria* × *ananassa*). This study emphasizes the importance of identifying the genotypes most resilient to low water availability. Experimental trials involving reduced irrigation levels were set up to identify genotypes with a greater capacity to increase fruit quality and maintain fruit yield. Reduced water conditions positively influenced strawberry fruit quality, exhibiting improved citric acid, soluble solids, and color brightness linked to decreased water use, while firmness remained stable. Notably, the total phenolic content was most affected by stress, indicating strong antioxidant responses. With these interesting variations in fruit quality came a different response in plant yield. Plants belonging to the Lauretta and AN15,07,53 cultivars maintained a 98% fruit yield when grown under WS1 conditions. While the yield for the Francesca cultivar increased by 10% under the stressed WS1 conditions in comparison to the control conditions, water stress in the WS2 treatment caused a strong reduction in yield in all three genotypes. Overall, the findings emphasize the importance of identifying for each new cultivar the most appropriate water regime in order to amplify the quality of the fruit, thus maintaining high production standards and saving water.

## 1. Introduction

The biggest ecological issues are currently water availability and distribution [1]. Many regions of the world are facing critical situations due to water scarcity, exacerbated by rising temperatures that increase evaporation rates and consequently raise the demand for water for crops [2]. Water scarcity affects more than 10% of Europe, particularly the southern regions, such as Italy and Spain [3]. Agriculture is the largest consumer of water compared to other sectors, accounting for 70% of the total annual water withdrawal [4,5]. Drought is an abiotic stress that hampers normal plant physiological activity, leading to physiological changes in plant structure, such as stomatal closure, reduced photosynthesis, minimized evapotranspiration, reduced biomass, and, consequently, decreased fruit production [6]. In this scenario, drought conditions are expected to worsen due to climate change, with rising air temperatures and increased atmospheric CO_2_ levels altering rainfall patterns and distribution [7,8,9,10]. Due to this issue, numerous experiments on precision irrigation and the selection of the most drought-resistant plants have been conducted for many crop species worldwide [11,12]. Among the tested crops, berry plants have gained attention due to the rising interest in them on the market [13,14]. Among berry plants, strawberries (*Fragaria* × *ananassa*) have been observed to have the highest demand, with 95.6 × 10^3^ tons produced globally in 2022 [15]. Strawberries contain high concentrations of molecules with antioxidant activity, such as vitamins, polyphenols, and phenolic acids, qualifying them as bio-functional foods [16]. Genetic diversity has been the basis for strawberry improvement because it is still an essential source of important traits that allow plants to adapt to climatic changes. In general, crops’ wild relatives exhibit valuable traits not maintained among cultivars, and these traits are critical for increasing adaptation to diverse environmental conditions [17]. Strawberry backcrossing programs of *F.* × *ananassa* combined with the two wild strawberry species (*F. virginiana* or *F. chiloensis)* are fundamental for generating new pre-breeding material carrying new essential traits [18]. Genomic studies on different breeding populations are now making progress in identifying adaptive and horticultural traits of high importance for genetically improving strawberry plants through the identification of new resilient cultivars [19,20]. Numerous studies have highlighted the critical need to investigate water-saving methods to optimize resource use [21,22] and examine changes in fruit quality. In strawberry cultivation, it is crucial to define the most appropriate irrigation regime, depending on the cultivar, the environment, and the cultivation system, based on the concept of returning evapotranspiration water from the plant. Determining the precise amount of water consumed by strawberries is challenging due to significant variations caused by genotypes and environmental conditions. However, earlier research [23] reported that in soil-based farming systems in the southeastern Marche region, water usage averages around 1120 m³/ha under standard conditions, with a water use efficiency (WUE) of approximately 37%. Tracking water used in conventional farming systems has proven to be complex, as precise measurements are difficult to maintain. Conversely, soilless farming systems are considered more adaptable, allowing for optimized resource management throughout the cultivation process. According to [24], WUE in soilless cultivation typically ranges between 31% and 41%, reflecting an improvement in water use flexibility and efficiency compared to traditional methods. The controlled water deficit approach is currently considered essential in order to be able to guarantee the correct use of water, with possible savings, maintaining yield and increasing quality. Marcellini et al. [23] revealed a significant gap in farmers’ knowledge of irrigation management, suggesting that water can be conserved without compromising production or quality. However, few studies have described the changes in bio-functional compound composition in strawberries from plants subjected to reduced irrigation. Our two-year study aimed, firstly, to investigate the productive, qualitative, and nutritional profiles of fruits harvested from strawberry genotypes of the UNIVPM breeding program; evaluate how the tested plants responded to different irrigation rates; and identify the most performant genotypes to use as parental materials. Secondarily, we tried to better understand how water shortages influence the bio-functional molecules accumulated in the fruits during cultivation and how dietary antioxidant compounds are correlated with each other.

## 2. Results and Discussion

### 2.1. Environmental Parameter

The water consumption of the plants was taken into consideration for two autumn production cycles: from 12 September to 7 November 2022, and from 9th September to 4th November 2023. The data are expressed in % of water volume content (%wvc), as shown in Figure 1. The plants at treatment WW maintained wvc around 44%, those at WS1 maintained wvc around 30%, and those at WS2 reached a level of humidity below 20%. Finally, the total water consumption reached 216 L for WW (100%), 160 L for WS1 (75%), and 130 L (60%).

### 2.2. Productive Parameters

We assessed the plants’ marketable yield (MY), the average fruit weight (AFW), and water use efficiency (WUE). Both the genotype and the treatment showed a crucial influence on fruit yield/plant and average fruit weight (Table 1), as investigated in other experiments [23]. At full irrigation (WW), the selection AN15,07,53 displayed the highest MY (185.52 g/plant), followed by ‘Lauretta’ (166.04 g/plant) and ‘Francesca’ (91.66 g/plant), as shown in Table 2. At WS1 for all genotypes, the MY was not statistically different from the WW treatment. For example, AN15,07,53 reached a marketable yield of 181.76 g/plant, ‘Lauretta’ reached 164.09 g/plant, and ‘Francesca’ reached 100.33 g/plant. The critical lack of production at WS2 seemed to be too high in comparison to the other treatments, with 91.33 g/plant for AN15,07,53, 85.47 g/plant for ‘Lauretta’, and 56.9 g/plant for ‘Francesca’. The investigation by Ödemiş et al., 2020 [25], revealed a similar trend. Gao et al., 2021 [26], explained the lack of production by the decrease in the plant’s photosynthetic activity. Regarding the AFW at full irrigation, the cultivar ‘Francesca’ showed the largest size, 14.11 g/fruit, while ‘Lauretta’ and the selection AN15,07,53 reached, respectively, 10.93 g/fruit and 10.09 g/fruit. Similarly to the loss of MY, the WS2 treatment exhibited the lowest values regarding the fruit weight, with 6.61 g/fruit for AN15,07,53, 8.77 g/fruit for ‘Lauretta’, and 8.1 g/fruit for ‘Francesca’. WUE had a remarkable role, and it was obtained from the ratio between kg of fruit production and the m^3^ of water supplied through irrigation. The WW did not prove to be the most efficient treatment, with a lack of efficiency in comparison with WS1 of about 25% in AN15,07,53, 33% in ‘Francesca’, and 26% in ‘Lauretta’. WS1 turned out to be the most efficient irrigation. The genotype AN15,07,53 was revealed to be the most efficient, producing 34 kg/m^3^; followed by ‘Lauretta’, with 30 kg/m^3^ of fruits; and ‘Francesca’, with 18.81 kg/m^3^ of fruits. WS2 had the most inefficient irrigation, reaching a loss of fruit production of 39% in AN15,07,53, 31% in ‘Lauretta’, and 36% in ‘Francesca’.

### 2.3. Sensorial Quality

The sensory quality of strawberry fruit is crucial, as it is directly perceived by the consumer and influences the commercial value of these fruits. Organoleptic quality is typically evaluated based on key parameters such as appearance, flavor, and firmness, which are indicators of fruit ripeness. These characteristics are primarily influenced by genotype and cultivation conditions. Both the genotype and the treatment broadly contribute to the expression of the fruit’s color, firmness, °Brix, and titratable acidity (Table 3). Specifically (Table 4), at WW AN15,07,53 showed the highest brightness (L*) in comparison to the other genotypes, with an average value of 40.16 L* compared to ‘Lauretta’ (37.42 L*) and ‘Francesca’ (36.27 L*). Moreover, the Chroma value, which is derived from the color dimension coordinates ‘a’ and ‘b’ using the formula [(a^2^ + b^2^)]^1/2^ to measure color saturation, was included as part of the organoleptic evaluation. It showed no visible differences in AN15,07,53 and ‘Francesca’ at full irrigation, with, respectively, 51.33 and 53.46 values. On the contrary, Lauretta’ exhibited a minor intensity in all the treatments. Concerning the fruits’ firmness, the results did not prove high differences between the genotypes at WW; in fact, AN15,07,53 reached 456.36 g/cm^2^, ‘Francesca’ reached 433.63 g/cm^2^, and ’Lauretta’ reached 422.72 g/cm^2^. Concerning the soluble solids (SS) of the fruits, in WW conditions, the AN15,07,53 genotypes exhibited 7.3 °Brix, ‘Francesca’ exhibited 7.6 °Brix, and ‘Lauretta’ exhibited 7 °Brix. The citric acid (CA) at WW was estimated at around 0.63% for AN15,07,53, 0.75% for ‘Francesca’, and 0.7% for ‘Lauretta’. WS2 resulted in a rise in fruit acidity. Comparing the treatments, we can conclude the following considerations. The reduction in water to WS1 and, more evidently, to WS2 implied a rise in the final fruit brightness in all genotypes. This may be attributed to the reduced canopy of the plants under water shortages, which likely leads to decreased light penetration within the canopy and subsequently affects the pigmentation of the fruits. [27,28]. For example, concerning the chroma values, a noticeable distinction was evident between WW and WS2 (e.g., the chroma of ‘Lauretta’ fruits at WW was 47.49, instead of at WS2, where it reached 48.35). Regarding the firmness of the fruits, the shortening of the irrigation volumes implied a hardening of the fruit consistency (e.g., in the WS1 condition, AN15,07,53 reached 473.63 g/cm^2^, ‘Francesca’ reached 470.9 g/cm^2^, and ‘Lauretta’ reached 455.45 g/cm^2^, from a consistency at WW of 456.36 g/cm^2^ for AN,15,07,53, 433.63 g/cm^2^ for ‘Francesca’, and 422.72 g/cm^2^ for ‘Lauretta’. A notable observation was the flattening of sweetness across all genotypes at WW, along with a general tendency to increase the °Brix content in the fruits from plants treated under WS2 conditions. Many studies revealed a similar tendency, entailing the sugar concentration as a response to drought conditions [10,16,29]. Some stimulating considerations may be given to the relationship between SS and drought tolerance since there is a positive correlation between the irrigation supply and the accumulation of soluble solids in many studies. The relation between the soluble solids concentration and the susceptibility to biotic stressors was positively described [30] concerning the *Botrytis cinerea* infection. Some considerations about the abiotic stressors and SSC of fruits may be interesting for future investigations. CA also exhibited correspondence between increased acidity and water reduction, as noticed in other studies [10,31]. As well as for the SS, the trend of the CA concentration suggests that different irrigation supplies may lead to different variations in respiratory metabolisms and biochemical pathways in citric acid regulation. In fact, in our study, the differences between WW and WS1 are similar; however, under WS2 conditions, the ‘Francesca’ plants showed a higher concentration of acidity in the fruits (e.g., WS2 in AN15,07,53 was about 0.75%; in ‘Francesca’, it was about 1.01%; and in ‘Lauretta’, it was about 0.96%). In contrast, other studies [32] demonstrated opposite behaviors for the citric acid concentration, showing a reduction in fruit acidity in plants subjected to drought conditions. In conclusion, at WS2, the plants exhibited brighter, sweeter, and sharper fruits but with irreversibly compromised fruit yield. The WS1 treatment resulted in a slight improvement in the fruit’s sweetness and acidity without compromising yield. Therefore, WS1 enabled 25% savings of water for irrigation, maintaining an appreciable fruit quality.

### 2.4. Nutritional Quality

Behind the scale-up of berry fruit consumption in recent decades, there is the consumer’s perception of the healthiness provided by the habitual consumption of these fruits, and generally by vegetable food products, as part of a common diet [33,34,35,36,37]. The antioxidant action provided by the bio-functional molecules present in strawberry fruits demonstrates effective radical scavenging activity against reactive oxygen species (ROS). [38,39]. The key molecules are vitamins, such as ascorbic acid (Asc.A.), folate (Fol.), flavonoids, and mainly anthocyanins, with the preponderant presence of perlargonidine-3-glucoside (Pel-3-gluc), cyanidine 3-glucoside (Cya.3-gluc), pelargonidine-3-rutinoside (Pel.3-rut), and phenolic compounds, such as chlorogenic acid (Chl.A.), ellagic acid (Ell.A.), and caffeic acid (Caff.A.) [38,40]. As highlighted by the experimentation (Table 5), the genotype has a consistent role in defining all the bio-functional molecules [41,42]. The irrigation treatment, too, showed great influence on the appearance of the following compounds, except for chlorogenic acid and caffeic acid, which appeared to be more correlated to the cultivar than to the treatment. Specifically, the genotype had a relevant influence on all the analyzed parameters (Table 6). In detail, under WW conditions, the cultivar ‘Lauretta’ exhibited the highest polyphenol cumulation (TPH), with107.7 mg GA/100 g; followed by ‘Francesca’, with 103.8 mg GA/100 g; and AN15,07,53 with 80.1 mg GA/100 g. Concerning the fruits’ total antioxidant capacity (TAC), ‘Lauretta’ exhibited the best performance, with 334.3 mg TROLOX eq/100 g; followed by ‘Francesca’, with 297.6 mg TROLOX eq/100 g; and AN15,07,53, at 252.01 mg TROLOX eq/100 g. Under WW conditions, the anthocyanin compounds (ACY) were most prevalent in ‘Lauretta’ with, respectively, 21.5 mg of Pel-3-gluc, 0.75 mg of Cya-gluc, and 1.97 mg of Pel-3-rut. Concerning the Pel-3-gluc, AN15,07,53 and ‘Francesca’, exhibited remarkable values with, respectively, 18 mg/100 g and 16 mg/100 g. Interestingly, the values of the Asc.A. content in ‘Francesca’ and ‘Lauretta’ with WW irrigation reached, respectively, 32.38 mg and 30.19 mg, in contrast with the selection, which reached 20.1 mg/100 g. A similar trend was observed in the FOL fruit content, where the cultivars exhibited the highest levels, with approximately 38.4 μg/100 g for ‘Francesca’ and 30.1 μg/100 g for ‘Lauretta,’ while the fruit of AN15, 07, 53 contained 28.6 μg/100 g. Finally, Chl.A. and Ell.A. are the most common phenolic acids present in fruits. ‘Francesca’ proved to be the most performant, with 16.7 mg (Chl.A.) and 9.1 mg (Ell.A.). ‘Lauretta’ and AN15,07,53 showed similar values, with, respectively, 9.3 mg/100 g and 9.4 mg/100 g for Chl.A. and 8.6 mg/100 g and 8.5 mg/100 g for Ell.A. The reduced irrigation entailed some changes in the fruits’ compound accumulation. The TPH analysis demonstrated a general trend across all genotypes, showing an increase in polyphenol fruit content with reduced water restitution (e.g., ‘Lauretta’ from 107.7 mg/100 g at WW to 122 mg/100 g at WS2, ‘Francesca’ from 103.5 mg/100 g to 127.5 mg/100 g, and AN15,07,53 from 80.1 mg/100 g to 91.3 mg/100 g). Other studies [43,44] reported similar results, indicating that the increased content of polyphenols is induced in response to the application of water stress. For all the genotypes, the TAC resulted in stability in all the treatments without relevant differences. Fruits of AN15,07,53 from plants under WS1 showed a small decrease in the total antioxidant capacity (237 mg TROLOX/100 g) compared to WW (252.1 mg TROLOX/100 g). On the contrary, fruits of ‘Francesca’ and ‘Lauretta’ showed the best performance at WS1 conditions, with, respectively, 322.5 mg TROLOX/100 g and 365.4 mg TROLOX/100 g. Concerning the anthocyanin content, Pel-3-gluc exhibited a small increase in response to stress conditions in all the genotypes (e.g., AN15,07,53 from 18 mg/100 g at WW to 19.3 mg/100 g at WS2 conditions). The overall trend showed a positive outcome for preserving anthocyanin content under WS1 conditions compared to WW because it allows water to be conserved while maintaining a significant level of anthocyanins in the fruit (e.g., for Pel-3-gluc, AN15,07,53 from 18 mg/100 g at WW to 17.9 in WS1 conditions, ’Francesca’ from 16 mg/100 g to 17.9 mg/100 g, and ‘Lauretta’ from 21.5 mg/100 g to 22.6 mg/100 g), as similarly found in other studies [45]. The relation between the fruit Asc.a. content and the treatment showed a small rise among the WW and WS1 in all genotypes. AN15,07,53 reached 20.1 mg/100 g at WW and 22.17 mg/100 g at WS1; Lauretta’ reached 30.19 mg/100 g at WW and 32.67 mg/100 g at WS1; and ‘Francesca’ reached 32.38 mg/100 g at WW and 32.76 mg/100 g at WS1. The Fol. content showed small differences only in ‘Francesca’ fruit (from 38.4 mg/100 g at WW and 40.6 mg/100 g at WS1) and AN15,07,53 (28.6 mg/100 g at WW and 27.2 mg/100 g at WS1), highlighting an overall trend to uphold the folic acid derivates independently via the given treatment. Chl.A., Caff.A., and Ell.A. did not show differences and remained constant between treatments without statistical significance. The only exception was Ell.A. in ‘Francesca’ under WS2 conditions, with 10.4 mg/100 g, in contrast to WW and WS1. The response to drought conditions involves several changes in plant physiology. Nonetheless, WS1 demonstrated an adequate preservation of the molecules without compromising other crucial parameters, like the fruits’ production. Furthermore, the cultivars stand with higher quality in comparison to the selection, and ‘Lauretta’ displayed a higher efficiency in terms of water use compared to the others. Finally, we set up a Pearson correlational analysis (Figure 2), providing the positive or negative relationship among all the antioxidant compounds. The map takes into consideration the relations among the molecules analyzed in all possible conditions (WW, WS1, and WS2) to create a coherent trend among the molecule’s concentration. The correlational analysis provided a brief, remarkable positive relationship among different molecules. In fact, narrow correlations were found among TAC-TPH and anthocyanins–FOL due to the active mechanism of free radicals scavenging [46]. As expected, the phenolic acids likewise were strictly related. On the other hand, the opposite trend was evident in the relation between Caff.A. and both Asc.A. and Pel.3-gluc and Pel.3-rut and Chl.A. Interestingly, the positive correlation among TPH and all the other compounds analyzed indicates the fact that the Folin-Ciocalteau assay is an analytical procedure still valid today, offering a reliable representation of the antioxidant profile of our fruit or vegetables [47,48].

## 3. Materials and Methods

This two-year experiment was conducted in the experimental greenhouse of the Università Politecnica delle Marche in Ancona (Italy). We tested the following three June bear strawberry genotypes from the UNIVPM breeding program: ‘Lauretta’, ‘Francesca’, and AN15,07,53 (‘Jonica’ × ‘Romina’). Generally, in soil farming, Francesca’ produced fruits with an excellent nutritional profile, ‘Lauretta’ exhibited a great compromise among healthy phytochemicals and an appreciable production yield, AN15,07,53 demonstrated great productivity and a set of promising dietary molecules [49]. For two autumn production cultivation cycles, mini-tray plants were planted on 9th September 2022 and 12th September 2023. Both the productions lasted until 3rd November 2022 and 9th November 2023, respectively.

### 3.1. Experimental Design and Irrigation Scheduling

The experimental irrigation started at the flowering bud’s appearance, at stage 55 on the BBCH scale [50], and ended on the last harvest date, at stage 89 on the BBCH scale [50]. For these experiments, a split-plot design was arranged for both years. Specifically, 90 plants were tested (36 plants for genotype). Each block corresponds to a water treatment (WW, WS1, and WS2, described further below), where the genotypes were replicated. The substrate used in these experiments is a mix of peat moss, coconut fiber, silicate, and eco-fiber (a vegetal material derived from renewable material) from a well-known company. The substrate was analyzed (Table 7) to fix the physicochemical and hydraulic properties. Three irrigation treatments were implemented based on the pF values outlined in the table below, representing the logarithmic factor (logarithm base 10) of the matric potential measured in centimeters of water and expressed as the water volume content (wvc), WW (pF1 at 45% wvc), WS1 (pF1-pF1.7 about 30% wvc), and WS2 (with some water shortage points, below pF2, about 21% wvc). Finally, the total water consumption reached 216 L for WW (100%), 160 L for WS1 (75%), and 130 L (60%). The water volume content was determined by adopting a TDR Fieldscout 150 (TDR Fieldscout 150, Plainfield, IL, USA) and a scale for the cross-check. The fertigation program, controlled by a Dosatron^®^ D8R (Dosatron SAS, Tresses, France), was applied for all the treatments. The average temperature and humidity inside the greenhouse were measured using the Testo175H1 sensor (Testo175H1, Lenzkirch, Germany), and both reached, respectively, 24 °C and 73%UR. The physiological status of the plant was measured using the LCi Portable Photosynthesis (LCi Portable Photosynthesis, Hoddesdon, United Kingdom) system, which confirmed the plant’s water status in terms of evapotranspiration.

### 3.2. Production Evaluation

The fruits, not rotted, misshaped, or not smaller than Ø < 15 mm, were harvested throughout the fruitification period. All the fruits were weighed to evaluate marketable yield per plant (total marketable yield/plants’ number) and average fruit weight (total marketable yield/number of harvested fruits). Finally, water use efficiency (total yield/m^3^ water) was determined. The fruits were collected on different days every year, specifically, 7th, 10th, 13th, 18th, and 28th of October and 3rd of November 2022 and 6th, 10th, 13th, 18th, 20th, 25th, and 31st of October 2023. Then, the harvested fruits were divided for organoleptic analyses and nutritional analysis.

### 3.3. Strawberry Fruit Quality

#### 3.3.1. Organoleptic Quality

The fruits were immediately analyzed using the colorimeter CR 400, Konica Minolta, Tokyo, Japan, measuring two points on opposite sides of each fruit using CIELAB values (L*, a*, b*). Then, the samples were frozen at −18 °C until the soluble solids (SS) and titratable acidity analysis using a digital refractometer (PR-101ATAGO, Tokyo, Japan) and the titrating burette, respectively. The TA was calculated as mEQ of NaOH per 100 g of fresh weight (FW) as follows: we added 10 g of distilled water and a few droplets of bromothymol blue (pH indicator) with a 0.1 N NaOH solution of 10 g of strawberry juice as the base [51].

#### 3.3.2. Bioactive Compounds

The strawberry fruits were stored at −18 °C in plastic bags after the harvesting, and 30 undamaged fruits were selected by each harvest day to avoid unwanted oxidative reactions. The frozen fruits were subjected to three different extraction procedures depending on the analytical procedure followed, as described [52]. Shortly, five fruits from the thirty were chosen, chopped, and weighed: 10 g was designated for the methanolic extract suitable for detection of antioxidant capacity (TAC), polyphenols (TPH), anthocyanins, and phenolic acids (Ph.A.); 1 g for detection of vitamin C (Asc.A.); and 2 g for extracting folates (Fol.) (folic acid derivates). Specifically, both TAC and TPH were evaluated through spectrophotometric analyses using a Shimadzu UV-1800 UV/Visible Scanning Spectrophotometer, while Acy, Ph.A., Asc.A., and Fol. were evaluated through liquid chromatography, the HPLC system (Jasco PU-2089 plus), and a Jasco UV-2070 plus ultraviolet (UV) detector (Jasco, Easton, MD, USA). TAC was measured through the 2,20-azino-bis (3-ethylbenzothiazoline-6-sulphonic acid) assay [53,54]. The TPH was calculated using the Folin–Ciocalteu reagent method [55]. The anthocyanins content, specifically for pelargonidin 3-glucoside, pelargonidin 3-rutinoside, and cyanidin 3-rutinoside, was analyzed by following the method outlined by Fredericks et al. [56]. The phenolic acids analyzed were chlorogenic, caffeic acid, and ellagic acid, as described by Schieber et al. [57]. Asc.A. was extracted and analyzed in accordance with the method described [58]. Finally, Fol., as derivates of folic acid, was extracted according to [59].

### 3.4. Statistical Analysis

The results are presented as the average values for the years 2022 and 2023 with the standard error and were subjected to a two-way analysis of variance (ANOVA) at 95% and 99% confidence levels. Significant differences were calculated according to Fisher’s LSD test, and differences at *p* < 0.05 were significant. Statistical analyses were performed using Statistica 7 software (StatSoft, TIBCO Software, Palo Alto, CA, USA). The Pearson correlation map was composed with JMP Pro 14.3 software.

## 4. Conclusions

Improving drought resistance in berry plants is a challenge for plant breeders due to the severity of this abiotic stress. Drought’s effects on plants’ metabolism have been widely described in many research studies. Nevertheless, little information is currently available on the relationship between the bio-functional compounds of strawberry fruits and the irrigation volumes supplied. The findings provided us with critical insights. Briefly, the water use efficiency highlighted the WS1 treatment as the most convenient irrigation due to the 25% water savings in comparison to full irrigation. At WS1, the WUE increased by 25% for AN15,07,53, 33% for ‘Francesca’, and 26% for ‘Lauretta’. The genotype AN15,07,53 responded in the most efficient way, producing 34 kg of fruits using 1 m^3^ of water; followed by ‘Lauretta’ WS1, with 30 kg of fruits; and ‘Francesca’, with 18.81 kg of fruits. The WS2 treatment is not optimal due to the critical lack of production in all genotypes, accounting for a lack of 39% in AN15,07,53, 31% in ‘Lauretta’, and 36% in ‘Francesca’. The organoleptic quality of the fruits showed an increase in SS, CA, and color brightness in response to the water deficit. Concerning the bio-functional compounds, TPH proved to be the most affected compound in terms of the water deficit. Specifically, AN15, 07, 53 showed an increase of 2% under WS1 and 12.2% under WS2; ‘Francesca’ demonstrated a rise of 15.7% under WS1 and 18.8% under WS2; and ‘Lauretta’ displayed an increase of 3% under WS1 and 15% under WS2. Under normal conditions, ‘Lauretta’ consistently demonstrated the highest TAC and Acy. Under reduced irrigation (WS1), ‘Lauretta’ exhibited minimal effects on critical compounds such as Acy and Asc.A. Furthermore, this genotype demonstrated yield stability in response to reduced water availability, suggesting that ‘Lauretta’ could be an important parent for breeding programs. Ultimately, the sensory and nutritional quality of the fruits is primarily influenced by the genotype [60], which also plays a crucial role in enhancing resilience to environmental stress conditions, including both biotic and abiotic stresses such as water scarcity. Certain molecular mechanisms are essential for improving plant resilience to these stress factors. This study confirms that polyphenol metabolism is directly involved in conferring improved resistance to water stress [33]. Furthermore, other compounds were evidently influenced by the shortening of the water supply, as well as Asc.a. and phenolic acids, even if in a nonsignificant manner. The effective integration of genomic analyses to identify the genes associated with the most influential polyphenols in drought resistance could be vital for optimizing future strawberry breeding programs.

## Figures and Tables

**Figure 1 plants-14-00286-f001:**
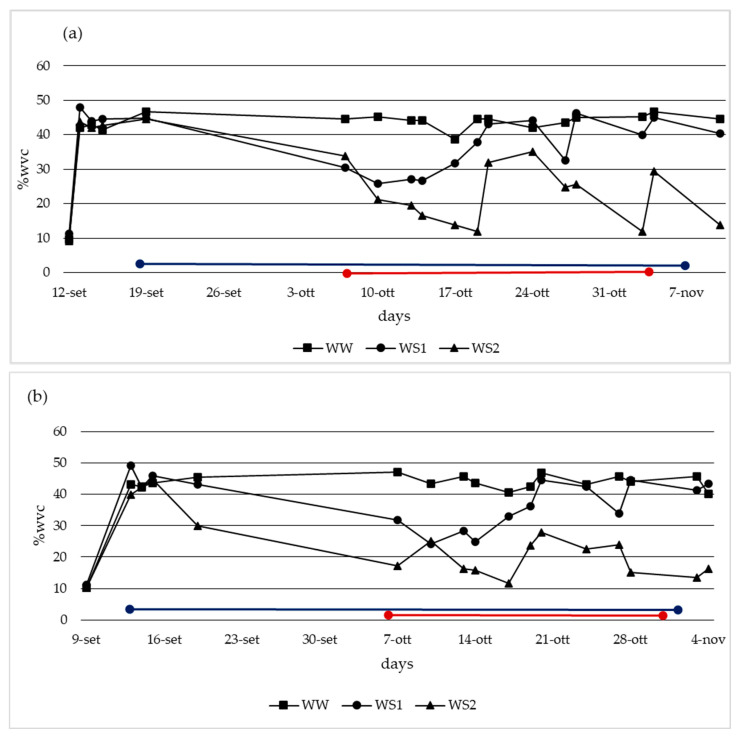
Water volume content (%wvc) of the three different treatments in 2022 (**a**) and 2023 (**b**). Blue line represents diversified irrigation periods; red line represents harvesting period.

**Figure 2 plants-14-00286-f002:**
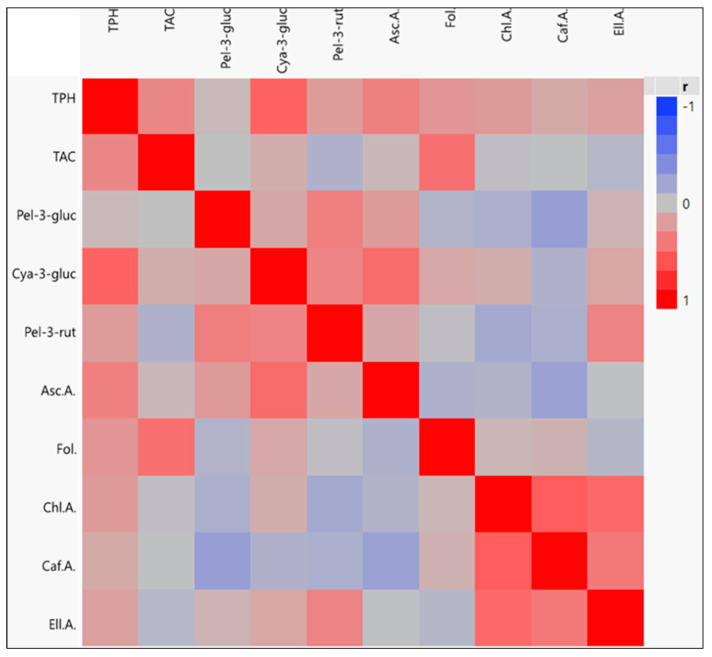
Heatmap of Pearson correlation coefficient of all evaluated traits in the three genotypes at three irrigation levels (WW 100%, WS1 75%, and WS2 56%). The right axis exhibits positive and negative correlations presented in red and blue colors, respectively, to the color scale of total phenolics (TPH), total antioxidant capacity (TAC), pelargonidine 3-glucoside (Pel.3-gluc), cyanidine 3-glucoside (Cya.3-gluc), pelargonidine-3-rutinoside (Pel.3-rut), ascorbic acid (Asc.a.), folates (Fol), chlorogenic acid (Chl.a.), ellagic acid (Ell.a.), and caffeic acid (Caff.a.).

**Table 1 plants-14-00286-t001:** Three-way analysis of variance (ANOVA) for marketable yield per plant (MY) and average fruit weight (AFW). The interaction among parameters was considered strongly significant with *p* values < 0.01 (**), slightly significant with *p* values < 0.05 (*), and non-significant with *p* values > 0.05 (NS).

Parameters	MY	AFW
Genotype	*	**
Year	NS	NS
Treatments	**	**
Genotype × Year	NS	NS
Genotype × Treatments	NS	NS
Year × Treatments	NS	NS
Genotype × Year × Treatments	NS	NS

**Table 2 plants-14-00286-t002:** Two-way ANOVA effect of different irrigation levels on the mean values (2022–2023) of marketable yield per plant (MY), average fruit weight (AFW) ± standard error is reported, and water use efficiency (WUE). Values with the same lowercase letter for the same parameter were not statistically different for Fisher’s LSD test (*p* < 0.05).

Treatment	Genotype	MY (g/Plant)	AFW (g/Fruit)	WUE (kg MY/m^3^ Water)
WW	AN15,07,53	185.52 ± 10.5 a	10.09 ± 2.68 bc	25.76
WS1	AN15,07,53	181.76 ± 9.67 a	8.48 ± 3.32 cd	34.08
WS2	AN15,07,53	91.33 ± 4.65 bcd	6.61 ± 2.6 e	21.07
WW	Francesca	91.66 ± 5.93 bcd	14.11 ± 3.48 a	12.73
WS1	Francesca	100.33 ± 8.11 bc	12.05 ± 4.66 ab	18.811
WS2	Francesca	56.9 ± 8.17 d	8.77 ± 4.19 cd	13.13
WW	Lauretta	166.04 ± 7.01 ab	10.93 ± 2.96 bc	23.06
WS1	Lauretta	164.09 ± 6.55 ab	9.41 ± 3.1 bcd	30.766
WS2	Lauretta	85.47 ± 7.08 bcd	8.1 ± 3.07 d	19.72

**Table 3 plants-14-00286-t003:** Three-way analysis of variance (ANOVA) for brightness (L*), chroma, firmness (F), soluble solids (SS), and citric acid content (CA). The interaction among parameters was considered strongly significant, with *p* values < 0.01 (**), slightly significant with *p* values < 0.05 (*), and non-significant with *p* values > 0.05 (NS).

Parameters	L*	Chroma	F	SS	CA
Genotype	**	**	**	*	**
Year	NS	NS	NS	NS	NS
Treatments	**	**	**	*	**
Genotype × Year	NS	NS	NS	NS	NS
Genotype × Treatments	**	**	NS	NS	NS
Year × Treatments	NS	NS	NS	NS	NS
Genotype × Year × Treatments	NS	NS	NS	NS	NS

**Table 4 plants-14-00286-t004:** Two-way ANOVA effect of different irrigation levels on the two years mean values (2022–2023) of brightness (L*), chroma, soluble solids (SS), firmness (F), and citric acid content (CA) ± standard error (2022–2023) are reported. Values with the same lowercase letter for the same parameter were not statistically different for Fisher’s LSD test (*p* < 0.05).

Treatment	Genotype	L*	Chroma	F	°Brix	CA
WW	AN15.07,53	40.16 ± 0.36 bc	51.33 ± 0.48 c	456.36 ± 3.02 ab	7.3 ± 0.27 bc	0.63 ± 0.032 e
WS1	AN15,07,53	40.52 ± 0.38 ab	52.03 ± 0.40 bc	473.63 ± 2.44 a	7.6 ± 0.54 bc	0.62 ± 0.01 e
WS2	AN15,07,53	41.18 ± 0.35 a	52.37 ± 0.54 bc	488.18 ± 2.81 a	7.9 ± 0.64 bc	0.75 ± 0.03 d
WW	Francesca	36.27 ± 0.26 h	53.46 ± 0.42 bc	433.63 ± 1.53 b	7.6 ± 0.14 bc	0.75 ± 0.05 d
WS1	Francesca	38.57 ± 0.33 de	53.51 ± 0.21 bc	470.9 ± 1.79 a	8.3 ± 0.27 ab	0.78 ± 0.03 cd
WS2	Francesca	39.47 ± 0.37 cd	55.29 ± 0.30 a	471.81 ± 1.61 a	9.1 ± 0.21 a	1.01 ± 0.03 a
WW	Lauretta	37.42 ± 0.28 fg	47.49 ± 0.37 e	422.72 ± 1.68 b	7 ± 0.69 c	0.79 ± 0.02 cd
WS1	Lauretta	37.79 ± 0.25 ef	49.68 ± 0.30 d	454.54 ± 3.02 ab	7.5 ± 0.23 bc	0.88 ± 0.03 bc
WS2	Lauretta	38.87 ± 0.29 de	48.35 ± 0.32 de	455.45 ± 2.7 ab	8 ± 0.24 bc	0.96 ± 0.02 ab

**Table 5 plants-14-00286-t005:** Three-way analysis of variance (ANOVA) of total phenolics (TPH), total antioxidant capacity (TAC), pelargonidine 3-glucoside (Pel.3-gluc), cyanidine 3-glucoside (Cya.3-gluc), pelargonidine-3-rutinoside (Pel.3-rut), ascorbic acid (Asc.a.), folates (Fol.), chlorogenic acid (Chl.a.), ellagic acid (Ell.a.), and caffeic acid (Caff.a.). The parameter interaction was considered strongly significant with *p* values < 0.01 (*), slightly significant with *p* values < 0.05 (**), and non-significant with *p* values > 0.05 (N.S.).

Parameters	TPH	TAC	Pel.3-gluc	Cya.3-gluc	Pel.3-rut
Genotype	**	**	**	**	**
Year	N.S.	*	**	*	N.S.
Treatment	**	*	*	**	**
Treatment × Year	**	**	N.S.	**	N.S.
Treatment × Genotype	**	*	N.S.	**	N.S.
Year × Genotype	*	**	**	N.S.	**
Treatment × Year × Genotype	**	**	**	**	**
**Parameters**	**Asc.a.**	**Fol.**	**Chl.ac.**	**Caf.ac.**	**Ell.ac.**
Genotype	**	**	**	**	**
Year	*	*	N.S.	N.S.	*
Treatment	**	*	N.S.	N.S.	*
Treatment × Year	N.S.	**	**	N.S.	N.S.
Treatment × Genotype	*	**	N.S.	**	**
Year × Genotype	**	**	N.S.	N.S.	**
Treatment × Year × Genotype	N.S.	**	**	**	**

**Table 6 plants-14-00286-t006:** Two-way ANOVA Effect of different irrigation levels on the two years’ (2022–2023) mean values of total phenolics (TPH), total antioxidant capacity (TAC), pelargonidine 3-glucoside (Pel.3-gluc), cyanidine 3-glucoside (Cya.3-gluc), pelargonidine-3-rutinoside (Pel.3-rut), ascorbic acid (Asc.a.), folates (Fol), chlorogenic acid (Chl.a.), ellagic acid (Ell.a.), and caffeic acid (Caff.a.) ± standard error. Values with the same lowercase letter for the same parameter were not statistically different for Fisher’s LSD test (*p* < 0.05).

Treatment	Genotype	TPH (mg GA/100 g)	TAC (mg TROLOX/100 g)	Pel-3-gluc (mg/100 g)	Cya-3-gluc (mg/100 g)	Pel-3-rut (mg/100 g)
WW	AN15,07,53	80.1 ± 1.21 d	252.1 ± 11.57 bc	18 ± 0.89 bcd	0.61 ± 0.01 c	1.08 ± 0.32 b
WS1	AN15,07,53	81.7 ± 0.99 d	237 ± 18.07 c	17.9 ± 0.97 cd	0.75 ± 0.005 c	1.38 ± 0.048 b
WS2	AN15,07,53	91.3 ± 1.02 cd	243.6 ± 19.49 c	19.3 ± 0.95 bcd	0.73 ± 0.009 c	1.37 ± 0.057 b
WW	Francesca	103.5 ± 0.62 b	297.6 ± 13.81 abc	16 ± 0.52 d	0.86 ± 0.01 b	1.59 ± 0.05 b
WS1	Francesca	122.8 ± 1.24 a	322.5 ± 20.88 abc	17.9 ± 1.23 cd	0.82 ± 0.02 b	1.46 ± 0.27 b
WS2	Francesca	127.5 ± 0.76 a	297.2 ± 14.56 abc	16.7 ± 0.87 d	0.9 ± 0.01 b	1.43 ± 0.08 ab
WW	Lauretta	107.7 ± 0.77 bc	334 ± 30.37 abc	21.5 ± 0.78 abc	0.75 ± 0.03 b	1.97 ± 0.2 ab
WS1	Lauretta	103 ± 1.23 bc	365.4 ± 29.13 a	22.6 ± 1.5 ab	0.84 ± 0.03 b	2.52 ± 0.26 ab
WS2	Lauretta	122 ± 1.33 a	314.5 ± 10.87 abc	25.4 ± 1.36 a	0.93 ± 0.03 a	3.30 ± 0.23 a
**Treatment**	**Genotype**	**Asc.A.** **(mg/100 g)**	**Fol. (μg/100 g)**	**Chl.A.** **(mg/100 g)**	**Caff.A.** **(mg/100 g)**	**Ell.A.** **(mg/100 g)**
WW	AN15,07,53	20.1 ± 0.33 d	28.6 ± 1.38 bc	9.4 ± 0.47 b	0.8 ± 0.01 bc	8.5 ± 0.08 b
WS1	AN15,07,53	22.17 ± 0.33 d	27.2 ± 0.82 c	8.7 ± 0.42 b	0.9 ± 0.11 ab	8.3 ± 0.2 b
WS2	AN15,07,53	22.98 ± 0.31 d	29.9 ± 0.83 bc	9.1 ± 0.47 b	0.8 ± 0.02 abc	8.6 ± 0.19 b
WW	Francesca	32.38 ± 0.61 ab	38.4 ± 2.56 ab	16.7 ± 1.23 a	0.9 ± 0.03 abc	9.1 ± 0.16 b
WS1	Francesca	32.76 ± 0.55 ab	40 ± 6.41 a	17 ± 1.76 a	1 ± 0.04 ab	8.8 ± 0.16 b
WS2	Francesca	32.8 ± 0.42 ab	32 ± 1.06 bc	16.1 ± 1.8 a	1 ± 0.05 a	10.4 ± 0.38 a
WW	Lauretta	30.19 ± 0.39 bc	30.1 ± 0.98 bc	9.3 ± 0.59 b	0.7 ± 0.02 cd	8.6 ± 0.18 b
WS1	Lauretta	32.67 ± 0.36 ab	29.4 ± 1.52 bc	7 ± 0.57 b	0.5 ± 0.009 d	8.9 ± 0.27 b
WS2	Lauretta	34.79 ± 0.38 a	29.7 ± 1.6 bc	7.7 ± 0.29 b	0.6 ± 0.02 cd	8.9 ± 0.32 b

**Table 7 plants-14-00286-t007:** Substrate analysis. The values represented show the physical and chemical parameters of the growing media.

Parameter	Unit of Measurement	Standard
Air volume pF1	50.50% *v*/*v*	UNI EN 13041:2012
Water volume pF1	45.08% *v*/*v*	UNI EN 13041:2012
Air volume pF1,7	71.39% *v*/*v*	UNI EN 13041:2012
Water volume pF1,7	24.19% *v*/*v*	UNI EN 13041:2012
Air volume pF2	74.15% *v*/*v*	UNI EN 13041:2012
Water volume pF2	21.43% *v*/*v*	UNI EN 13041:2012
Total Porosity	95.58%	UNI EN 13041:2012
Ph	6.2 pH units	UNI EN 13040:2008 + UNI EN 13037:2012
EC	43 mS/m	UNI EN 13040:2008 + UNI EN 13037:2012
N-NH4	19.96 mg/L	UNI EN 13652:2001
N-NO3	93.71 mg/L	UNI EN 13652:2001
P	38.84 mg/L	UNI EN 13652:2001
Ca	18.12 mg/L	UNI EN 13652:2001
Mg	11.80 mg/L	UNI EN 13652:2001
K	441.47 mg/L	UNI EN 13652:2001
Na	105.45 mg/L	UNI EN 13652:2001

## Data Availability

The original contributions presented in this study are included in the article. Further inquiries can be directed to the corresponding author.

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
