# Peer review of "Yield and Sensorial and Nutritional Quality of Strawberry (*Fragaria* × *ananassa* Duch.) Fruits from Plants Grown Under Different Amounts of Irrigation in Soilless Cultivation"

_plants, 2025, doi:10.3390/plants14020286_

Round 1

Reviewer 1 Report

Comments and Suggestions for Authors

The manuscript contains relevant research results, but the presentation of the data does not fully reflect the value of the work completed. I propose a major revision of the manuscript from three aspects:

a)      Eliminate general statements without supporting data

b)     Add more details to the description of the experiments

c)      Check wording and use the appropriate English expressions in sentences, improve the flow and readability of the text

Specific comments and proposals:

1.      Lines 15-17: “The study highlights the importance of understanding the water retention capacity of the growing substrate…” The same statement is repeated in the Conclusions. In fact, water retention experiments were not described in the manuscript. The statement is generally true, but the reader expects that different substrates were compared in the experiments.

2.      Please insert space between the numerical value and the unit symbol: e.g., write 185.52 g/plant instead of 185.52g/plant,

3.      Line 91: Please indicate that WW denotes a treatment. (…treatment WW)

4.      Line 95: The figure quality is low, and the font size is small. Please rearrange the figures horizontally and increase font size and resolution.

5.      Line 95: The figure caption should be Figure 1.

6.      Line 101: Use square brackets for the reference [23].

7.      Line 141: The color of the samples was determined in the CIELAB (L*a*b*) color space. Only two out of the three color coordinates were reported. I suggest briefly explaining how chroma is defined and why the hue value was omitted from the analysis.

8.      Line 189: The Table 3 caption refers to CA as the citric acid content. The header of Table 3 shows TA in the last column. Please correct.

9.      Line 192: There is Table 4 and Table 4a-4b in the manuscript. This is confusing. I suggest presenting Table 4a-4b as two separate tables like Table 5 and Table 6.

10.   Line 207: Please correct the typo: chlorogenic cacid

11.   Line 286: I suggest splitting Table 5a-5b into two separate tables.

12.   Line 311: Please be more verbose at the first mention of an abbreviation. For example, …at stage 55 on the BBCH-scale.

13.   Line 316: I suggest specifying the source of eco-fibers.

14.   Line 329: Please write a more detailed caption for Table 7. The table contains information not only about the substrate but also shows the composition of the nutrient solution. The Table caption should explain the meaning of tq in the middle column. Please use the notation of liter consistently throughout the paper. In the table lower case l denotes the liter, in line 323 capital L is used.

15.   Lines 377-381: General sentences without information.  

16.   Line 381: “The genotype response to the water reduction” – probably an unfinished sentence.

17.   Line 383: There is no need to redefine water use efficiency in the Conclusions.

18.   Lines 393-396: another corrupted sentence

19.   Lines 411-413: Pathogens were not investigated; therefore, this sentence creates confusion in the Conclusion section.

20.   The Conclusions is mainly a repetition of the previously described results. It can be shortened and focused on the general learnings, quantifying, for example, the cost saving associated with an optimized irrigation schedule.

21. Line 42: Write COinstead of CO2

21.   Some references are missing: 11, 13, 15, please update the list of references

Comments on the Quality of English Language

Check wording and use the appropriate English expressions in sentences to improve the flow and readability of the text.

There are several corrupted sentences in the text, as highlighted above.

Author Response

Dear Reviewer,

I really appreciate your correction. I will change, and redefine the corrupted sentences in the text, to make it more digestable for the readers.

I will answer you point by point.

1)Lines 15-17: “The study highlights the importance of understanding the water retention capacity of the growing substrate…” The same statement is repeated in the Conclusions. In fact, water retention experiments were not described in the manuscript. The statement is generally true, but the reader expects that different substrates were compared in the experiments.

1) I substitute the sentence with 'the study emphasizes the importance to identify the most resilient genotypes to low water availability'

2)Please insert space between the numerical value and the unit symbol: e.g., write 185.52 g/plant instead of 185.52g/plant,

2)I inserted space between the numerical value and the unit symbol

3)   Line 91: Please indicate that WW denotes a treatment. (…treatment WW)

3) Thank you for the correction. Done

4) Line 95: The figure quality is low, and the font size is small. Please rearrange the figures horizontally and increase font size and resolution.

4) Thank you for the suggestion. I increased the size of the pictures, improving the quality of the images.

5)Line 95: The figure caption should be Figure 1.

5)Thank you. Done.

6) Line 101: Use square brackets for the reference [23].

6) Dear reviewer, thank you for the correction.

7) Line 141: The color of the samples was determined in the CIELAB (L*a*b*) color space. Only two out of the three color coordinates were reported. I suggest briefly explaining how chroma is defined and why the hue value was omitted from the analysis.

7)I added the sentence 'Chroma value or color saturation, obtained from the color dimension coordinates ‘a’ value and ‘b’ value, with the formula [(a2 + b2)]*1/2, was considered as part of the organoleptic evaluation'. The 'a' and 'b' values were omissed to highlight mostly the importance of *L and Chroma (as derivate of the relation among these two values), without prolonging the text getting it more digestable.

8)Line 189: The Table 3 caption refers to CA as the citric acid content. The header of Table 3 shows TA in the last column. Please correct.

8)Thank you for the correction, i corrected with CA

9)Line 192: There is Table 4 and Table 4a-4b in the manuscript. This is confusing. I suggest presenting Table 4a-4b as two separate tables like Table 5 and Table 6.

9)Thank you. I changed the tables' enumeration to to make the understanding clearer to the reader

10)Line 207: Please correct the typo: chlorogenic cacid

10)Thank you.

11) Line 286: I suggest splitting Table 5a-5b into two separate tables.

11)Thank you for the suggestion. I've done.

12) Line 311: Please be more verbose at the first mention of an abbreviation. For example, …at stage 55 on the BBCH-scale.

12)Thank you for the correction.

13)Line 316: I suggest specifying the source of eco-fibers.

13)Thank you for the suggesstion. Ecofiber is a material patented by the substrate company.

14) Line 329: Please write a more detailed caption for Table 7. The table contains information not only about the substrate but also shows the composition of the nutrient solution. The Table caption should explain the meaning of tq in the middle column. Please use the notation of liter consistently throughout the paper. In the table lower case l denotes the liter, in line 323 capital L is used.

14) Thank you for the suggestion. I decided to delete the term 'tq' that states for 'tal quale' in italian, or 'such as' in inglesh. I chenged also the quotation of the table. 

15) Lines 377-381: General sentences without information.  

15) Thank you, I deleted the sentence.

16) Line 381: “The genotype response to the water reduction” – probably an unfinished sentence.

16)Thank you, I completed the sentence.

17)Line 383: There is no need to redefine water use efficiency in the Conclusions.

17) Thank you for the suggestion, i deleted that part.

18)Lines 393-396: another corrupted sentence

18)Thank you, I deleted the sentence

19)Lines 411-413: Pathogens were not investigated; therefore, this sentence creates confusion in the Conclusion section.

19)Thank you, I deleted the sentence.

20) The Conclusions is mainly a repetition of the previously described results. It can be shortened and focused on the general learnings, quantifying, for example, the cost saving associated with an optimized irrigation schedule.

20)Thank you for the suggestion. I will try to optimize the conclusion's content

21)Line 42: Write CO2 instead of CO2

21)Thank you, I've done. 

22) Some references are missing: 11, 13, 15, please update the list of references

22)Thank you, I've done. 

Reviewer 2 Report

Comments and Suggestions for Authors

The results of your investigation are very interesting for other researchers or genetic improvement. In line 35, do you have any information what parts of Europe suffer more from water scarcity.  In line 47 you mean the world's production or? Some parts of the manuscript are marked (lines 101-122, 208-236 and 252-268) to ask for the inclusion of other authors who worked with these varieties or under similar conditions. In Material and methods I am missing the values of the average temperature and relative humidy inside the greenhouse. In the Reference list the references 11, 13 and 15 are missing.

Author Response

1) The results of your investigation are very interesting for other researchers or genetic improvement.

1) Dear Reviewer, I'm so grateful to you  for your interest in my research topic.

2) In line 35, do you have any information what parts of Europe suffer more from water scarcity. 

2) Dear Reviewer, I included other information: "Water scarcity affects more than 10% of Europe, particularly in the southern regions, such as Italy and Spain(etc.)"

3) In line 47 you mean the world's production or?

3) Dear reviewer, thank you for the clarification. Yes, I was referring to the global production. This the sentence I added in the manuscript "Within berry plants, strawberries (Fragaria x ananassa) have shown the highest demand, producing globally 95.6 x 103 tons in 2022".

4) Some parts of the manuscript are marked (lines 101-122, 208-236 and 252-268) to ask for the inclusion of other authors who worked with these varieties or under similar conditions.

4) Dear reviewer, I added some quotations in the lines 101-122, 208-236 and 252-268, to increment the references of the people who worked in the same field.

5) In Material and methods I am missing the values of the average temperature and relative humidy inside the greenhouse.

5) Dear reviewer, thank you for the correction. I added the values you answered. I used a TESTO175H1 sensor for both the experiments. 

6) In the Reference list the references 11, 13 and 15 are missing.

6) Dear reviewer, thank you for the observation. I was in mistake for the references list.  

Round 2

Reviewer 1 Report

Comments and Suggestions for Authors

Line 172: Please correct the formula by showing 1/2 as an upper index without the asterisk (*). 

Author Response

1)Line 172: Please correct the formula by showing 1/2 as an upper index without the asterisk (*). 

1) Dear reviewer, thank you for the correction.